# The Assessment of School Lunches in the Form of Food Packs during the COVID-19 Pandemic in Latvia

**DOI:** 10.3390/children9101459

**Published:** 2022-09-23

**Authors:** Ilze Beitane, Sandra Iriste

**Affiliations:** Department of Nutrition, Faculty of Food Technology, Latvia University of Life Sciences and Technologies, LV-3004 Jelgava, Latvia

**Keywords:** school lunch, food packs, pandemic, pupils, composition

## Abstract

During the pandemic, Latvian schools switched to remote learning which required looking for solutions to provide state-funded school lunches for pupils at home. The aim of study was to analyse the type of support received by pupils for provision of school lunches, the compliance of the composition of food packs with Latvian healthy diet recommendations and parental assessment of the food packs received. With the help of the questionnaire data on the composition of food packs, parental assessment was obtained by interviewing 1495 parents of pupils in grades 1–4 (age 6–11 years). The composition of food packs was evaluated in accordance with the recommendations for a healthy diet. Food packs were the main choice for the provision of school lunches in all regions (90.70%). The emphasis in the food packs was on protein-rich products like canned meat and meat products (93.36%) and milk (91.37%). 81.71% of food packs contained both vegetables and fruits. Food packs covered basic needs but improvements would be needed. The parents appreciated the support received, 90% of them rated it as positive/partially positive. In the event of a pandemic recurrence, the state would need to work with food producers to provide smaller packaging of products for food packs to ensure food diversity.

## 1. Introduction

Children are the ones who will determine the quality of life of the next generation, so it is important to ensure a favourable environment for them, including a healthy diet. In Europe, the increasing prevalence of childhood obesity [1], which affects life expectancy, life quality and health in adulthood, shows that the goal is not being met. In Latvia, overweight and obesity rates among pupils in the period from 2010 to 2018 have seen an upward trend in all age groups; moreover, the highest prevalence in 2018 was for 11-years old boys (30%) and the lowest—for 15-years old girls (15%) [2]. This means taking every opportunity to improve the nutrition and increase physical activity for pupils. One of the most efficient ways to influence the nutrition of school-aged children is a healthy school lunch, which can have a positive effect on both pupils’ behaviour patterns and educational achievement [3,4] as well as an impact on the pupils’ eating behaviour [5]. The advantage of the schools is that they cover a great number of pupils, can have an impact on the content of school food and can tackle the obesity issue among pupils [5,6,7]. O’Brien et al. [8] point out that schools have been identified as a key place to influence the healthy eating habits of pupils. School nutrition in Latvia is regulated by Cabinet Regulation of the Republic of Latvia No. 172 “Regulations regarding nutritional norms for educatees of educational institutions, clients of social care and social rehabilitation institutions and patients of medical treatment institutions” [9], which determines the provision of nutrients (protein, fat and carbohydrates) and energy to pupils, indications of the products to be included, and a list of products prohibited in the diet of pupils at school, for instance, sugar confectionary and flour pastry products containing partially hydrogenated vegetable fats, non-alcoholic drinks with added caffeine and/or amino acids. The list of not-recommended/prohibited products was created with the goal of restricting the choice of unhealthy products in the pupils’ diet, given that the choice of products is determined by taste and convenience, so the unavailability of less healthy foods in the school cafeteria/canteen promotes healthy food consumption [10]. Improving the quality of school meals and limiting choices to healthier options might be the right way to achieve healthy food consumptions among pupils [11]. The Latvian Ministry of Health’s recommendations for healthy eating in children include fresh or cooked vegetables, fruits, berries at every meal; the menu should include products from different food groups (grain products and potatoes, fruits and vegetables, milk and dairy products, meat, fish as well as other protein-source products, fat and oil); fish should be included in the diet at least 2 times a week; the consumption of sugar and salt should be limited [12].

Until the outbreak of the COVID-19 pandemic, all pupils from 1st to 4th grade (age 6–11 years) in Latvian schools were provided with state-funded school lunches (a warm and healthy meal) [13]. With the onset of the pandemic and the implementation of remote learning, the question of how to provide school lunches for pupils had to be addressed. Approaches to the issue varied from country to country. For instance, in Israel, School Feeding Program was suspended during pandemic [14]. A similar situation was seen in US, where long-term school closures during pandemic denied access to free and reduced-price school meals for millions of pupils [15]. This could be the cause of malnutrition among pupils as studies have shown that children are at greater risk of malnutrition during the summer period when schools are closed [16], and pupils have unhealthier eating habits, which in turn increases the risk of weight gain [17]. During remote learning, school lunches in Estonia were provided with food packs, in Finland—prepared takeaway meals, and in Sweden—lunch boxes [18]. A national voucher schema was implemented in the UK, which allowed families with school-aged children to buy products weekly worth 15.00 GBP in supermarkets [19]. In Latvia, food packs were the simplest and safest form of state support for families with pupils [20]. In order to facilitate the development of food packs for schools/municipalities, the recommendation of the Ministry of Health of the Republic of Latvia “Recommendations to municipalities for the provision of food packs to pupils” was developed [21], specifying one week’s food packs of pupils from 1st to 4th grade should include the following products: 900 g of fruit and vegetables, 600 g of milk and dairy products, 600 g of grain products and potatoes, 500 g of protein- source products and 90 g of fat (oil). In addition, prototypes of food packs were created as part of a national project, in line with the Latvian recommendations of a healthy diet and the nutrient and energy value requirements for pupils [22].

The above-mentioned activities (recommendations, prototypes of food packs and received food packs) generated a great response, some parents expressed negative opinions about food packs on social media, while the mass media covered it more widely. Therefore, it was important to understand whether the decision taken at the national level in favour of food packs can be considered a successful solution, as studies on the experiences of other countries [14,15,18,19] showed that approaches were different in provision of school lunches during remote learning.

In order to assess the support received by Latvian pupils in the form of food packs for the provision of school lunches during remote learning, the aim of the study was to analyse: (1) the type of support received by pupils for provision of state-funded school lunches; (2) the compliance of the composition of food packs with healthy diet recommendations; (3) parental assessment of the food packs received.

## 2. Materials and Methods

### 2.1. Data & Variables

This paper presents part of the outcomes of the study, which took place in Latvian schools between April and June 2021. The questionnaire was sent to 356 schools’ administrations with a request to send it to parents via the website—*e-klase*, which is Latvia’s electronic school management system. Schools were randomly selected across the country using data from Ministry of Education and Science on Latvian schools on the website—*skolu karte* (school map) [23].

The study followed the ethical standards recognized by the Latvian Academy of Sciences and the Latvian Council of Science [24]. All participants (parents) who completed the questionnaire in Google forms provided their written informed consent to participate in this study.

The questionnaire included 45 questions, of which five were closed-ended, two were open-ended, four were line-scale and the rest were semi-open-ended. The results of questionnaire were downloaded in Microsoft Excel 2013 file from the survey administration software Google forms and used for further analysis. The data were expressed in numbers and percentages, in addition, divided by regions (Figure 1) and in total. In the study six regions (Riga, Riga region, Vidzeme, Zemgale, Latgale, Kurzeme) were distinguished, one of which was the capital of Latvia—Riga because about half of the population of Latvia is concentrated there. Out of all 634 schools in Latvia, 136 are located in Riga [25].

A total of 6120 parents of pupils in grades 1–12 (age 6–19 years) from all Latvian regions were surveyed in the study, of which 1495 were parents of pupils in grades 1–4 (age 6–11 years), whose questionnaires were selected for this study. Finally, the authors analysed 1356 questionnaires, because that is how many pupils received food packs during the COVID-19 pandemic. Distribution of respondents by regions: Riga—364, Riga region—289, Vidzeme—141, Zemgale—284, Latgale—194, and Kurzeme—84. Cabinet Regulations of Republic of Latvia No. 172 “Regulations regarding nutritional norms for educatees of educational institutions, clients of social care and social rehabilitation institutions and patients of medical treatment institutions” and recommendations of the Ministry of Health of the Republic of Latvia “Recommendations to municipalities for the provision of food packs to pupils” were used to analyse the healthiness of food packs. The authors evaluated the obtained data on the composition of food packs with regulations and recommendations where the list of desirable and undesirable food products is indicated.

### 2.2. Statistical Analysis

The study used primary mathematical data processing methods to obtain descriptive statistics, visualization of the obtained data in graphs and charts. Secondary mathematical data processing methods were used to obtain inferential statistics in SPSS 21.0 (IBM Corp. Released 2020. IBM SPSS Statistics for Windows, Version 21.0. Armonk, NY: IBM Corp). Analysis of variance (ANOVA) was used to determine significant differences in the composition of food packs between regions while correlation analysis was used to evaluate the strength of correlation between parents’ positive evaluation of food packs and the food groups included in food packs.

## 3. Results

### 3.1. The Type of Support Received by Pupils for the Provision of School Lunches during the Pandemic

The way to provide school lunches during remote learning was not determined at the national level, the choice was left to each school. Only after a large number of schools chose food packs, the Ministry of Health quickly developed and approved recommendations on the desired composition of the food pack. However, this does not prevent the school from choosing a different approach. The approaches chosen by schools to provide support for the state-funded school lunch varied; however, most schools preferred food packs (Table 1).

Food packs were the main choice for the provision of the school lunch in all regions (between 76.36% in Kurzeme and 97.97% in Riga regions), while other approaches, such as gift cards or vouchers, money in the bank account etc. were uncommon, thereby only food packs were further analysed in the study.

### 3.2. The Compliance of the Composition of the Food Packs with Recommendations for a Healthy Diet

The actual composition of included products in food packs is shown in Table 2, without conducting a study on the quantities of food products.

The most frequently included product groups in food packs were canned meat or meat products (93.36%), milk (91.37%), fruits and vegetables (81.71%), eggs (79.06%) and grain products (78.39%). Surprisingly, bread was relatively rarely (48.38%) included in food packs and there were regional differences in the type of bread. In Riga and Zemgale, rye bread, bread with seeds or grains was the type most often included in food packs while in the Riga region, Vidzeme, Latgale and Kurzeme—white bread. Legumes were sufficiently often included in food packs (71.83%) as a source of protein while canned fish was the least frequently included product (36.95%). However, it should be noted that significant regional differences were observed for certain product groups: canned fish was included in Kurzeme in 7.14% of cases while in Riga—59.34% of cases; dairy products—in Riga, 30.77% of cases while in Latgale, 94.84% of cases; nuts, seeds, dried fruits—in Latgale, 6.18% of cases while in Riga, 92.58% of cases. Oil was included in 50% of cases and more depending from region, which was implemented by including one bottle of oil (rapeseed or sunflower) in the food pack once at a time. An essential fact was that the parents identified that sweets, such cookies, waffles and bars, were presented in the food packs at a higher frequency than bread, with the exception of parents in Vidzeme and Zemgale.

Evaluating the differences in the composition of food packs in the regions, the processing of data statistics showed that the *p*-value was 0.0989 < α = 0.1, thus, with a probability of 90%, it can be assumed that the composition of food packs (product groups included) depended on the region.

The compliance of the composition of food packs with the recommendations of Ministry of Health [12,21] is reflected in Table 3. The data shows the percentage of the food packs, which contained food products included in the recommendations.

In general, all groups of food products included in the recommendations were used in the design of the food packs. However, the presence of fruits and vegetables in food packs could be higher, the authors of study expected that fruits and vegetables would be included in the food packs in 90% and more cases, as they are one of the base of a healthy diet. In Latgale and Kurzeme, fruits and vegetables were least included in food packs (70.62% and 69.05% respectively). From the product group—milk and dairy products, most schools chose UHT milk due to the storage conditions, dairy products were chosen less frequently. Grain products such pasta, rice and buckwheat were included in food packs more often than bread, which should be evaluated as a positive fact from the point of view of healthy nutrition. Regarding protein-source products, the approaches were different, the most frequently included products were canned meat of meat products, followed by eggs or legumes and finally canned fish. From the product group—fat, schools included a bottle of oil (rapeseed or sunflower) in food packs once throughout the support period. However, some schools (50% in Kurzeme and 23.94% in Zemgale) chose not included this product group (fat) in the food pack.

### 3.3. Parental Assessment of the Food Packs Received

Overall, parental assessment of the food packs was positive (Table 4), as parents appreciated the support received and the food products were most often included in the diet of the whole family.

Assuming that a partially positive assessment is considered to be a good result, then in a total of 91.07% of cases it could be stated that the chosen approach in providing school lunches in the form of food packs was justified from the point of view of parents. Overall, less than 3.0% of parents gave a partially negative and negative assessment, indicating that the products included in food packs were of poor quality and inconsistent with the family’s diet, except in Zemgale, where the partially negative and negative rating was 3.87%, and Kurzeme, where there were no negative reviews.

The correlation analysis was used to assess whether there was strong correlation between the positive parental assessment of food packs and the food groups included in the food packs (Figure 2).

A strong correlation was observed for the following food groups: grain products, fruits and vegetables, canned meat and meat products, canned fish and fish products, eggs, legumes, milk, oil, nuts, seeds and dried fruits, sweets, which means that the presence of these products in food packs contributed to parental satisfaction ensuring a positive assessment of the food packs. A moderately strong correlation was shown for bread, while a weak correlation was found for dairy products.

## 4. Discussion

The choice of Latvian schools to provide school lunches with the help of food packs can be assessed in two ways: on one hand, it was relatively the easiest, fastest, and safest way to provide food. The products were taken already pre-packaged from producers or wholesalers, the schools simply designed the food packs. On the other hand, it made it difficult for schools to design food packs according to the recommendations, taking into account the nutritional needs of pupils. Existing food packaging was larger than necessary. This, in turn, prevented schools from ensuring food diversity in food packs, as the weekly cost of a food pack was 7.10 EUR (one school lunch was priced at 1.42 EUR per pupil). The quick reaction of the Ministry of Health to the current situation and development of recommendations for schools/municipalities to design food packs was seen as a positive [20].

In evaluating the composition of food packs, it must be concluded that the emphasis was placed on protein-rich products, as the most common products included in food packs were canned meat and meat products, milk, eggs, and legumes. In the food group of canned meat and meat products, parents most often referred to canned meat for complaints because such products did not fit into the family diet, and sausages, which were a popular meat product among children. The inclusion of canned meat in food packs can be assessed in different ways. On one hand, it is a source of protein, but most often high in fat as lean meat is not used in the production of canned meat, plus high salt content. In turn, the regulation of the Cabinet of Ministers [9] states that only lean meat should be used in the diet of pupils; meat products, including canned meat, should be included only if they contain at least 70% meat and that the salt content is less than 1.25 g per 100 g of meat product. When designing food packs, the emphasis was on the food prices rather than nutritional value. On the other hand, it should be noted that only products able to be kept at room temperature could be included in the list of products for food packs because no attention was paid to the food storage temperature during the designing storage and delivery of food packs. This meant that fresh meat or chicken fillets were not suitable for food packs, which explains the schools’ choice in favour of canned meat.

Regarding the choice of milk, UHT milk was included in food packs due to storage conditions as it can be stored unopened in room temperature. However, some parents indicated that the food packs included UHT milk with different flavours—strawberry, banana, chocolate, which have a sugar content about twice as high as unflavoured milk. Given that sugar consumption in the general population, including children and adolescents, exceeds the WHO recommendations [26], the choice of schools in favour of flavoured UHT milk could not be considered positive, as it is also contrary to the Latvian Ministry of Health’s recommendations for healthy eating in children [12]. In terms of dairy products, sour cream, yogurt and cottage cheese were included in food packs.

Legumes were included in 71.83% of food packs, mainly canned beans because it is cheap but nutritious product. For canned beans, attention should be paid to the salt content, which is on average less than 1 g per 100 g of product. However, given that food packs also contained canned meat with an average salt content of 1.5 g per 100 g of product, it is important to take into account that the daily recommendation of less than 5 g of salt per day [27] is not exceeded.

A healthy diet is based on a daily intake of at least 5 portions of fruits and vegetables [12,27], but the study found that only 81.71% of food packs contained both vegetables and fruits. In Latvia, the intake of fruits and vegetables among adolescents (11–15) is extremely low, as less than one third consume at least one fruit and one vegetable every day (26.8% and 27.2%, respectively) [2]. Therefore, it would have been desirable for schools to include both fruits and vegetables in each food pack, as home availability of fruits and vegetables aids in their consumption [28]. For vegetables, carrots, beets and cabbage were most often included in food packs, while for fruits, apples were most common. Unfortunately, the principle of diversity for fruits and vegetables was not respected.

Grain products, including bread, are an important source of daily energy. The results of the study showed that 78.39% of food packs contained grain products such as rice, buckwheat and pasta. Parents most often stated that they received pasta in food packs, and some of them added that the quality of pasta was low—it was cheap, and the family did not eat that kind of pasta. The parents also pointed out that the composition of food packs did not change significantly from week to week, so there ended up being a stock of pasta, rice or buckwheat, because the family was reluctant to eat the same product every day. Bread was included in 48.38% of food packs, where part of food packs contained either white bread or rye/grain/seed bread, but there were also food packs that contained two types of breads: white and rye/grain/seed bread. The addition of bread in food packs could be explained by the easiest way to fulfil the recommendation to provide 600 g of grain products [21]. The recommendations of healthy nutrition indicate that the emphasis should have been on whole grain products [12,27], and, therefore, the choice of schools in favour of white bread in Vidzeme and Zemgale regions (49.64% and 48.94%, respectively) is not understandable. It would be advisable to replace white bread with more nutritious bread, given that food packs contained white instead of brown rice, pasta instead of whole grain pasta. Unfortunately, grain products included in food packs do not promote fibre intake, as whole grain products, along with fruits, vegetables, legumes and nuts, have been shown to improve fibre intake among children [29]. The inclusion of nuts, seeds and dried fruits in food packs differed essentially between regions, which could be explained by the additional support of different municipalities in providing school lunches. Nuts, seeds and dried fruits are relatively expensive products, which are practically impossible to include in food packs at a cost of 7.10 EUR. Riga, as a capital city, is the richest municipality in Latvia providing additional support to schools, so 92.58% of food packs in Riga contained nuts, seeds and dried fruits.

The presence of sweets in food packs can also be assessed in two ways. On one hand, children have a high need for energy due to their growth and development processes, but on the other hand, there was limited mobility during the COVID-19 pandemic, which necessitated a reduced energy intake. However, studies showed that people ate more and dietary patterns were unhealthy [30,31]. Therefore, the fact that 55.60% of food packs contained sweets could be considered a negative aspect. In addition, parents pointed out items, such as cheap cookies and waffles or pasties filled with jam, as nutritionally poor sweets.

In general, schools have taken the existing recommendations into account when designing the food packs but the study showed that there are differences between regions which could be due to the various interpretations of recommendations, additional local municipality support, and the school’s target for food packs. Despite the shortcomings of the food packs in terms of healthy eating recommendations, parental assessment confirmed that the food packs’ overarching goal of providing food support to pupils during the pandemic was achieved, as more than 90% of parents rated it as positive or partially positive. The parents appreciated the support received, however they identified shortcomings of the food packs: the lack of food diversity from one week to the next; the unacceptable presence of certain products, such as canned meat, which strays from family eating habits.

This study has several strengths, such as a large number of respondents who represented all regions of Latvia, allowing to get a general insight of the country as a whole. It also shows the willingness of parents and administrators to improve the overall situation by actively participating in the study. The results of study made it possible to evaluate the choice of food packs as an option of providing school lunches. The study showed the issues that should be solved in the future in case of repeated pandemic, for instance, smaller packages of food products to ensure the diversity of products included in the food packs.

The study has a few limitations. First, respondents were grouped by region of residence, income level and social status were not taken into account, which could influence parents’ assessment of the support they received. Second, parents may misreport the food products included in food packs as the study was carried out after the support was provided. Third, the study did not examine the amount of products included in food packs due the concerns that parents might give incorrect answers. The same type of products can have different package weights. Another limitation is that this study did not carry out calculations of the nutritional and energetic value of food packs in order to evaluate their compliance with nutritional norms due to a lack of information on the quantities of food packs.

## 5. Conclusions

Given the crisis situation in the country during the COVID-19 pandemic and sufficiently rapid state response to the change in the form of pupil lunch support, the food packs may be considered sufficiently successful support to families with pupils. The composition of food packs for school lunches covered basic needs, but improvements would be needed to clearly confirm it was in line with the Latvian recommendations for a healthy diet, e.g., more fruits and vegetables, healthier snacks like nuts, seeds and dried fruits instead of sweets, whole grain bread instead of white bread, lean canned meat.

In the event of a pandemic recurrence, the state, as the contracting authority, would need to work with food producers to provide smaller packaging for food, to design food packs in accordance with the recommendations, and to ensure food diversity and safety.

## Figures and Tables

**Figure 1 children-09-01459-f001:**
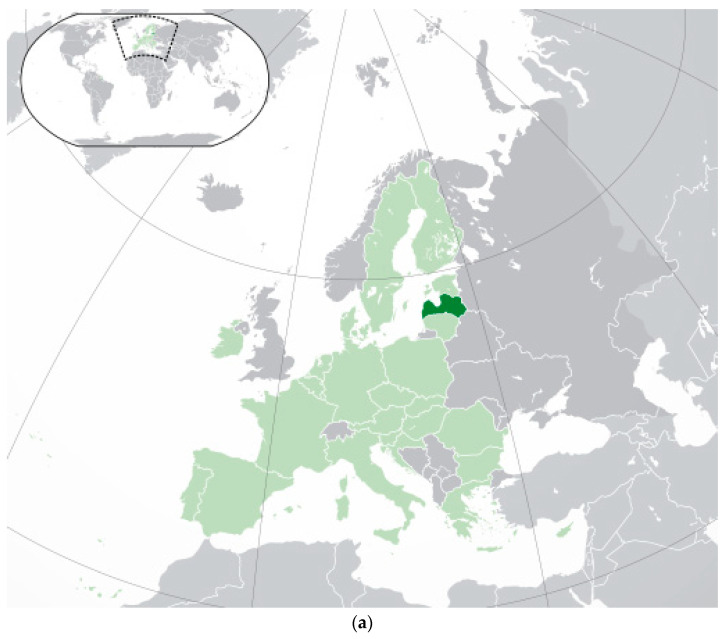
Maps: (**a**) Location of Latvia (dark green) by Nuclear Vacuum—This W3C-unspecified vector image was created with Inkscape, CC BY-SA 3.0, https://commons.wikimedia.org/w/index.php?curid=8105197 (accessed on 16 August 2022); (**b**) Regions of Latvia by Central Statistical Bureau of Latvia—Central Statistical Bureau of Latvia, CC BY-SA 3.0, https://commons.wikimedia.org/w/index.php?curid=18527500 (accessed on 16 August 2022).

**Figure 2 children-09-01459-f002:**
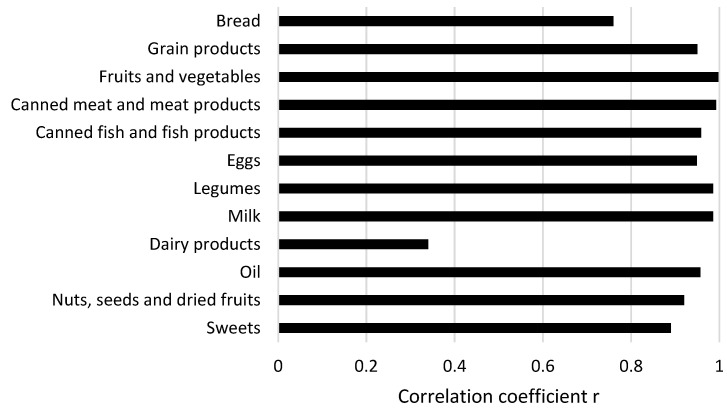
The correlation between parents’ positive assessment of the food packs and the product groups included in the food packs.

**Table 1 children-09-01459-t001:** The type of support for pupils to provide school lunches during the pandemic by Latvian region.

The Type of Support	Riga	Riga Region	Vidzeme	Zemgale	Latgale	Kurzeme	Total
Food packs	364	289	141	284	194	84	1356
	95.54%	97.97%	89.81%	86.32%	86.99%	76.36%	90.70%
Gift card or voucher	6	-	6	36	22	12	82
	1.57%	-	3.82%	10.94%	9.86%	10.91%	5.48%
Money in bank account	5	1	2	3	5	1	17
	1.31%	0.34%	1.27%	0.91%	2.24%	0.91%	1.14%
Take-away lunches	4	2	4	1	1	-	12
	1.05%	0.68%	2.55%	0.30%	0.45%	-	0.80%
Lunch delivered at home	1	-	3	4	-	9	17
	0.26%	-	1.91%	1.21%	-	8.18%	1.14%
Other	1	3	1	1	1	4	11
	0.26%	1.02%	0.64%	0.30%	0.45%	3.64	0.73%
Total	381	295	157	329	223	110	1495

**Table 2 children-09-01459-t002:** Composition of food packs by Latvian region.

Product Groups	Riga	Riga Region	Vidzeme	Zemgale	Latgale	Kurzeme	Total
Bread	156	95	104	193	72	36	656
	42.86%	32.87	73.76%	67.96%	37.11%	42.86%	48.38%
Rye bread, bread with seeds or grains	115	69	49	164	17	17	431
	31.59%	23.87%	34.75%	57.75%	8.76%	20.24%	31.78%
White bread	50	76	70	139	43	23	401
	13.74%	26.30%	49.64%	48.94%	22.16%	27.38%	29.57%
Grain products (rice, buckwheat, pasta etc.)	314	180	110	222	175	62	1063
	86.26%	62.28%	78.01%	78.17%	90.21%	73.81%	78.39%
Fruits and vegetables ^1^	318	243	113	239	137	58	1108
	87.36%	84.08%	80.14%	84.15%	70.62%	69.05%	81.71%
Canned meat or meat products	352	282	125	260	177	70	1266
	96.70%	97.58%	88.65%	91.55%	91.24%	83.33%	93.36%
Canned fish	216	104	24	133	18	6	501
	59.34%	35.99%	17.02%	46.83%	9.28%	7.14%	36.95%
Eggs	315	181	111	224	177	64	1072
	86.54%	62.63%	78.72%	78.87%	91.24%	76.19%	79.06%
Legumes	264	222	95	210	139	44	974
	72.53%	76.82%	67.37%	73.94%	71.65%	52.38%	71.83%
Milk	361	283	122	231	176	66	1239
	99.17%	97.92%	86.52%	81.34%	90.72%	78.57%	91.37%
Dairy products	112	143	127	170	184	54	790
	30.77%	49.48%	90.07%	59.86%	94.84%	64.28%	58.26%
Oil	226	175	92	216	127	42	878
	62.09%	60.55%	65.25%	76.06%	65.46%	50.00%	64.75%
Nuts, seeds, dried fruits	337	165	58	153	12	34	759
	92.58%	57.09%	41.13%	53.87%	6.18%	40.48%	55.97%
Sweets	183	136	80	155	147	53	754
	50.27%	47.06%	56.74%	54.58%	75.77%	63.09%	55.60%

^1^ The food pack included both fruits and vegetables.

**Table 3 children-09-01459-t003:** The compliance of the composition of food packs with the recommendations, %.

Recommendation	Products	Riga	Riga Region	Vidzeme	Zemgale	Latgale	Kurzeme	Total
Fruits and vegetables	Fruits and vegetables	87.36	84.08	80.14	84.15	70.62	69.05	81.71
Milk and dairy products	Milk	99.17	97.92	86.52	81.34	90.72	78.57	91.37
	Dairy products	30.77	49.48	90.07	59.86	94.84	64.28	58.26
Grain products	Bread	42.86	32.87	73.76	67.96	37.11	42.86	48.38
	Grain products	86.26	62.28	78.01	78.17	90.21	73.81	78.39
Protein-source products	Canned meat or meat products	96.70	97.58	88.65	91.55	91.24	83.33	93.36
	Canned fish	59.34	35.99	17.02	46.83	9.28	7.14	36.95
	Legumes	72.53	76.82	67.37	73.94	71.65	52.38	71.83
	Eggs	86.54	62.63	78.72	78.87	91.24	76.19	79.06
Fat (oil)	Oil	62.09	60.55	65.25	76.06	65.46	50.00	64.75

**Table 4 children-09-01459-t004:** Parental assessment of the food packs.

Assessment	Riga	Riga Region	Vidzeme	Zemgale	Latgale	Kurzeme	Total
Positively	277	210	116	204	132	65	1004
	76.10%	72.66%	82.27%	71.83%	68.04%	77.38%	74.04%
Partially positive	53	55	17	54	37	15	231
	14.56%	19.03%	12.06%	19.01%	19.07%	17.86%	17.03%
Neutral	27	16	5	15	15	4	82
	7.42%	5.54%	3.55%	5.28%	7.73%	4.76%	6.05%
Partially negative	5	7	1	6	7	-	26
	1.37%	2.42%	0.71%	2.11%	3.61%	-	1.92%
Negatively	2	1	2	5	3	-	13
	0.55%	0.35%	1.42%	1.76%	1.55%	-	0.96%

## Data Availability

Not applicable.

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
