# Peer review of "The Assessment of School Lunches in the Form of Food Packs during the COVID-19 Pandemic in Latvia"

_children, 2022, doi:10.3390/children9101459_

Round 1

Reviewer 1 Report (New Reviewer)

Dear Authors; I found this work an interesting descriptive study to evaluate the food packs delivery program effectiveness during COVID19 pandemic in Latvia. It does need some serious extra work to arrive to publication quality. Regards. P.S.

[1] Writing:

1-1 Make sure the references are in MDPI format. For example years for articles are in bold format.

1-2 The "niche" of study in the introduction section is missing. What is missing in the related literature that justifies writing this paper ? (before line 66). Add a paragraph.

1-3 Section "2.Materials and Methods" need to be broken down into two subsections: "2.1. Data & Variables", and "2.2. Statistical Analysis".

1-4 In line 103, SPSS21.0 needs its reference! See: *IBM Corp. Released 2020. IBM SPSS Statistics for Windows, Version 21.0. Armonk, NY: IBM Corp

1-5 Add Map of Latvia location in Europe and with its regions mentioned in the Table 1 in one Figure to the paper. Your paper international readers deserve to know  where is Latvia and which regions in it are the topic of discussion  in the paper ! Use the following two maps-as a suggestion: 

https://en.wikipedia.org/wiki/Latvia 

https://en.wikipedia.org/wiki/Statistical_regions_of_Latvia

1-6 The "5.Conclusion" section is too much. Summarize it for overall 5-6 lines. Move the extra parts to the "4.Discussion" section.

[2] Statistical:

2-1 Do reported correlations in Figure.1 have p-values? Report them as adjacent column.

2-2 Line 145: p-value of what was 0.0989? Add it to the text.

Author Response

Dear Reviewer,

Thanks for all your comments and suggestions after reviewing our manuscript.

  1. Make sure the references are in MDPI format. For example years for articles are in bold format.

Author answer: The corrections were made in the publication.

  1. The "niche" of study in the introduction section is missing. What is missing in the related literature that justifies writing this paper? (before line 66). Add a paragraph.

Author answer: Authors added a paragraph to the publication.

  1. Section "2.Materials and Methods" need to be broken down into two subsections: "2.1. Data & Variables", and "2.2. Statistical Analysis".

Author answer: The corrections were made in the publication.

  1. In line 103, SPSS21.0 needs its reference! See: *IBM Corp. Released 2020. IBM SPSS Statistics for Windows, Version 21.0. Armonk, NY: IBM Corp

Author answer: The correction was made in the publication.

  1. Add Map of Latvia location in Europe and with its regions mentioned in the Table 1 in one Figure to the paper. Your paper international readers deserve to know  where is Latvia and which regions in it are the topic of discussion  in the paper ! Use the following two maps-as a suggestion: https://en.wikipedia.org/wiki/Latvia  and https://en.wikipedia.org/wiki/Statistical_regions_of_Latvia

Author answer: The corrections were made in the publication.

  1. The "5.Conclusion" section is too much. Summarize it for overall 5-6 lines. Move the extra parts to the "4.Discussion" section.

Author answer: The corrections were made in the publication.

  1. Do reported correlations in Figure.1 have p-values? Report them as adjacent column.

Author answer: The figure shows the correlation coefficient between parents’ positive assessment of the food packs and the product groups included in the food packs. An explanation of the correlation coefficient is given. P-value was not calculated.

  1. Line 145: p-value of what was 0.0989? Add it to the text.

Author answer: The correction was made in the publication.

Reviewer 2 Report (New Reviewer)

General

Authors of the manuscript cover important issue of state-funded school lunches provision during the COVID-19 pandemic and resultant remote learning. The topic is important and up-to-date and the obtained results are interesting.

However, I have serious concerns associated with one of the aims of the study, namely “analysis of the compliance of the composition of food packs with healthy diet recommendations”. Authors assess the compliance of meals with recommendations of the Latvian Ministry of Health “Recommendations to municipalities for the provision of food packs to pupils”. In these recommendations we read that food packs of pupils for one week should include different amounts of products, for example 900 g of  fruits and vegetables, 600 g of milk and dairy products etc. However, while analyzing food packs provided during remote learning Authors only assess the presence of specific food products in the packs, like bread, fruits and vegetables etc. However, they don’t analyze the amounts of products included in the food pack, so in fact they are not able to state that the meal is in compliance with Latvian recommendations. In order to properly answer to this aim of the study Authors should also include questions about typical serving size in the questionnaire.

Abstract

-Line 14 – please add brief information about sampling method used in the study.

-Line 16 – it needs clarification. What kind of recommendations do you mean here? Latvian ones?

-Line 19 – delete double space.

Introduction

-Lines 46-49 – please provide more details what kind of products are limited in Latvian schools according to the regulation.

-It would be more valuable if Authors cite more studies across Europe showing how the matter of providing school meals to pupils during lockdown was solved.

-Line 64 – once again, what kind of recommendations do you mean here? Please provide more information regarding these recommendations for pupils (what they include).

Materials and Methods

-Did you obtain the approval of Institutional Review Board? If so, give the number of the decision.

-Line 73 – how were schools chosen? Were they randomly selected? Did you sent questionnaire to schools from the whole country? You have to provide more details about sampling procedure.

-Line 75 – delete double space.

-Another question is whether you performed sample size calculation?

-Did you use any validated questionnaire or was it proprietary questionnaire?

Results

-Line 119 – I think that information about recommendations of Ministry of Health about food packs for pupils should be placed earlier in the manuscript, in introduction or M&M

-As recommendations about food packs for pupils show the amounts of different products

Discussion

-There are no strengths and limitations

References

-Authors should include more references to provide proper background for the manuscript.

Author Response

Dear Reviewer,

Thanks for all your comments and suggestions after reviewing our manuscript.

However, I have serious concerns associated with one of the aims of the study, namely “analysis of the compliance of the composition of food packs with healthy diet recommendations”. Authors assess the compliance of meals with recommendations of the Latvian Ministry of Health “Recommendations to municipalities for the provision of food packs to pupils”. In these recommendations we read that food packs of pupils for one week should include different amounts of products, for example 900 g of fruits and vegetables, 600 g of milk and dairy products etc. However, while analyzing food packs provided during remote learning Authors only assess the presence of specific food products in the packs, like bread, fruits and vegetables etc. However, they don’t analyze the amounts of products included in the food pack, so in fact they are not able to state that the meal is in compliance with Latvian recommendations. In order to properly answer to this aim of the study Authors should also include questions about typical serving size in the questionnaire.

Author comment: I agree with the reviewer that questions about product amounts in food packs were not included in the questionnaires because the authors were concerned about incorrect answers that would lower the reliability of the data, since the study was conducted after receiving the support. This means that the parents had to fill in the questionnaire based on the recall principle, which is one of the limitations of the study. But according to the authors' opinion, this does not prevent the overall assessment of the compliance of food packages with healthy nutrition recommendations and the recommendations developed by the Ministry of Health” Recommendations to municipalities for the provision of food packs to pupils”. The study assessed whether the mentioned food groups were included and what products were chosen from the product group, for example protein-source products - the choices are wide, but the most popular choice was canned meat.

Abstract

-Line 14 – please add brief information about sampling method used in the study.

Author answer: The abstract has a word limit, so it is not possible to add information about the sample method, it is indicated in the section Materials and methods.

-Line 16 – it needs clarification. What kind of recommendations do you mean here? Latvian ones?

Author answer: The correction was made in the publication.

-Line 19 – delete double space.

Author answer: The correction was made in the publication.

Introduction

-Lines 46-49 – please provide more details what kind of products are limited in Latvian schools according to the regulation.

Author answer: The corrections were made in the publication.

-It would be more valuable if Authors cite more studies across Europe showing how the matter of providing school meals to pupils during lockdown was solved.

Author answer: More studies are added in the publication.

-Line 64 – once again, what kind of recommendations do you mean here? Please provide more information regarding these recommendations for pupils (what they include).

Author answer: This is a reference to another study where prototypes of food packs were developed as part of the project. The publication has been supplemented with additional information on recommendations.

Materials and Methods

-Did you obtain the approval of Institutional Review Board? If so, give the number of the decision.

Author answer: The study followed the ethical standards recognized by the Latvian Academy of Sciences and the Latvian Council of Science (Latvian Academy of Science, Latvian Science Council. Code of ethics for scientists. Available online: https://lzp.gov.lv/wp-content/uploads/2020/10/Etikas_kodekss_LV.pdf). All participants (parents) who completed the questionnaire in Google forms provided their written informed consent to participate in this study.

-Line 73 – how were schools chosen? Were they randomly selected? Did you sent questionnaire to schools from the whole country? You have to provide more details about sampling procedure.

Author answer: The corrections were made in the publication.

-Line 75 – delete double space.

Author answer: The correction was made in the publication.

-Another question is whether you performed sample size calculation?

Author answer: The percentage distribution of product groups has been calculated for each region separately, because otherwise the regions could not be compared with each other due to the different number of respondents.

-Did you use any validated questionnaire or was it proprietary questionnaire?

Author answer: A questionnaire was developed for this study.The pilot study was conducted with 20 parents, approbating the questionnaire. The results of pilot study showed that all the questions in the questionnaire were clear and understandable. None of the questions required further clarification.

Results

-Line 119 – I think that information about recommendations of Ministry of Health about food packs for pupils should be placed earlier in the manuscript, in introduction or M&M

Author answer: The corrections were made in the publication.

-As recommendations about food packs for pupils show the amounts of different products

Author answer: Parents were asked about the products included in food packs without conducting a study on the quantities of food products, because this question would be difficult for the respondents to answer, and the reliability of the data would be low, since the same type of product can have different package weights.

Discussion

-There are no strengths and limitations

Author answer: The authors added a strengths and limitations paragraph to the publication.

-Authors should include more references to provide proper background for the manuscript.

Author answer: More references are added in the publication.

Reviewer 3 Report (New Reviewer)

The article addresses the issue of providing school meals to pupils during the pandemic. It is an interesting topic, considering the value of the school meal and the problems of childhood obesity that increase during the long periods in which children stay at home and do not go to school.

The article states that it has three objectives, the first of which is to analyze the type of support received by the children for provision of school lunches. This objective deserves to be deepened in the manuscript both in terms of the methodology with which it was addressed and in terms of results. In particular, in the results session, I would like to understand who decided the strategy adopted by the schools, if there were any national indications that schools could follow, the motivation for the prevalence of the “food packs” solution, the organization (In the abstract you mention the relationship with food producers (L 22), but the text of the manuscript speaks of wholesalers, L 212), the logistics and the costs of this strategy. Something of this type is mentioned in the discussion but, since there is no reference in the methodology and in the results, it is not clear where the information contained in the discussion and relating to this first objective comes from. I suggest adding the required information to point 3.1.

Also the objective number 2 needs a more development, in particular I would suggest to insert a table showing the main points of the Latvian diet recommendations and the compliance with the composition of the food packs in the 3.2 session.

Below I add some detailed indications.

Abstract

At line 15 please add the years to which the grades correspond.

There is an error at line 19, a dash before the percentage.

I would remove the word safety (L 23) because the connection between a smaller packaging and food safety is not explained in the manuscript.

Introduction

Please, insert references on the relations between children malnutrition and long period of school closure.

See, for example:

Huang, J., Barnidge, E. and Kim, Y., 2015, “Children receiving free or reduced-price school lunch have higher food insufficiency rates in summer”, The Journal of Nutrition, 145(9):2161–2168.

Franckle, R., Adler, R. and Davison, K., 2014, “Accelerated weight gain among children during summer versus school year and related racial/ethnic disparities: a systematic review”, Preventing Chronic Disease, 11:E101.

Lines 58-59, insert "in the world" or something like that.

Lines 62-63: There are good solutions also in US, see for example,

Kinsey, E.W., Hecht, A.A., Dunn, C.G., Levi, R., Read, M.A., Smith, C., Niesen, P., Seligman, H.K. and Hager, E.R., 2020, “School Closures During COVID-19: Opportunities for Innovation in Meal Service”, American Journal of Public Health, 110(11): 1635- 1643.

Please move the paragraph of lines 76-82 first, at line 54.

At line 82 please explain which are “the above mentioned activities”.

Material and methods

Please, give details of the eight sections mentioned at line 105.

Lines 121-122: I did not understand why the total (6,120 parents) does not match the number of the 1,495 questionnaires. Please, explain.

Author Response

Dear Reviewer,

Thanks for all your comments and suggestions after reviewing our manuscript.

  1. The article states that it has three objectives, the first of which is to analyze the type of support received by the children for provision of school lunches. This objective deserves to be deepened in the manuscript both in terms of the methodology with which it was addressed and in terms of results. In particular, in the results session, I would like to understand who decided the strategy adopted by the schools, if there were any national indications that schools could follow, the motivation for the prevalence of the “food packs” solution, the organization (In the abstract you mention the relationship with food producers (L 22), but the text of the manuscript speaks of wholesalers, L 212), the logistics and the costs of this strategy. Something of this type is mentioned in the discussion but, since there is no reference in the methodology and in the results, it is not clear where the information contained in the discussion and relating to this first objective comes from. I suggest adding the required information to point 3.1.

Author answer: The required information to point 3.1 is added.

  1. Also the objective number 2 needs a more development, in particular I would suggest to insert a table showing the main points of the Latvian diet recommendations and the compliance with the composition of the food packs in the 3.2 session.

Author answer: The Table is inserted and some information is added.

Abstract

At line 15 please add the years to which the grades correspond.

Author answer: The information is added.

There is an error at line 19, a dash before the percentage.

Author answer: The correction was made in the publication.

I would remove the word safety (L 23) because the connection between a smaller packaging and food safety is not explained in the manuscript.

Author answer: The correction was made in the publication.

Introduction

Please, insert references on the relations between children malnutrition and long period of school closure.

See, for example:

Huang, J., Barnidge, E. and Kim, Y., 2015, “Children receiving free or reduced-price school lunch have higher food insufficiency rates in summer”, The Journal of Nutrition, 145(9):2161–2168.

Franckle, R., Adler, R. and Davison, K., 2014, “Accelerated weight gain among children during summer versus school year and related racial/ethnic disparities: a systematic review”, Preventing Chronic Disease, 11:E101.

Author answer: The suggested references are added.

Lines 58-59, insert "in the world" or something like that.

Author answer: The correction was made in the publication.

Lines 62-63: There are good solutions also in US, see for example,

Kinsey, E.W., Hecht, A.A., Dunn, C.G., Levi, R., Read, M.A., Smith, C., Niesen, P., Seligman, H.K. and Hager, E.R., 2020, “School Closures During COVID-19: Opportunities for Innovation in Meal Service”, American Journal of Public Health, 110(11): 1635- 1643.

Author answer: Thank you for the suggestion. The reference was already contained in the publication.

Please move the paragraph of lines 76-82 first, at line 54.

Author answer: The correction was made in the publication.

At line 82 please explain which are “the above mentioned activities”.

Author answer: The correction was made in the publication.

Material and methods

Please, give details of the eight sections mentioned at line 105.

Author answer: The correction was made in the publication. The division of sections in the publication has been removed. Sections in this case were a technical solution offered by Google Forms to move to the next group of questions depending on the answer. For example, if the respondent answered that the support was money in the bank account, then the questions about the composition of the food packs were skipped.

Lines 121-122: I did not understand why the total (6,120 parents) does not match the number of the 1,495 questionnaires. Please, explain.

Author answer: The correction was made in the publication.

 A total of 6,120 parents of pupils in grades 1-12 (age 6-19 years) from all Latvian regions were surveyed in the study, of which 1,495 were parents of pupils in grades 1-4 (age 6-11 years), whose questionnaires were selected for this study.

Round 2

Reviewer 1 Report (New Reviewer)

Dear Authors; most of my concerns were addressed satisfactorily. Regards.

Author Response

Dear reviewer,

Thank you for your invested work and time in improving the publication.

Yours sincerely,

Ilze Beitane

Reviewer 2 Report (New Reviewer)

Authors have responded to majority of my concerns (especially associated with not assessing the serving sizes of products provided within food packs). However, one thing needs clarification - line 98 - how were schools selected to participate in the study? Do you have any national database of schools?

Author Response

Dear reviewer,

Thank you for your invested work and time in improving the publication.

The authors added additional information to the publication about school selection.

Schools were randomly selected across the country using data from Ministry of Education and Science on Latvian schools on the website – skolu karte (school map) [21].

Yours sincerely

Ilze Beitane

Reviewer 3 Report (New Reviewer)

Thanks for taking my suggestions into consideration.

This manuscript is a resubmission of an earlier submission. The following is a list of the peer review reports and author responses from that submission.

Round 1

Reviewer 1 Report

The manuscript entitled The assessment of school lunches in the form of food packs during the COVID-19 pandemic describes how Latvian schools dealt with the issue of providing school lunches to their pupils during the COVID-19 pandemic. The assessment of food packs in terms of compliance with national regulations and the opinion of pupils' parents was presented. Below please find my review along with   suggestions for this manuscript:

  1. The abstract does not clearly state what the purpose of the study was. Please complete this information very deeply.
  2. Please specify how many schools were included in the survey.
  3. Please explain why only students in 1st-4th grade were included in this survey since school lunches are provided in grades 1st-6th?
  4. Please complete this paper with a detailed description of the statistics used and a detailed presentation of the results of the statistical analysis.
  5. Figure 1. To maintain consistency in the order of the product groups included in the survey and shown in Figure 1, I would suggest reordering the groups in Figure 1.
  6. Chapter conclusions - line 244 - 246, I would suggest giving examples of better diversification of pupils' healthy diet in line with the recommendations, e.g. more fruit and vegetables etc. In my opinion, this chapter should also briefly mention the general opinion of parents about this food pack solution since finding out this opinion was one of the objectives of this work. For this purpose, it might be worth considering moving the last paragraph from the discussion section to the conclusions?
  7. The questionnaire should be added to the supplementary materials or to section Materials and Methods
  8. There is no clearly emphasized purpose and sense of the research undertaken, please describe what is the demand for such knowledge, why did the authors decide to take up this topic, please indicate how the obtained results will affect the future behavior of consumers, what will these results bring to science?

The presented results are not data of high scientific value for the food discipline, they provide knowledge that can be found on many internet portals/popular sources. Foods journal, which has IF over 5, publish articles with the broad scientific soundness, unfortunately, despite the fact that the reviewed article is interesting, the research it contains are very poor and provides only basic knowledge. Because the purpose of the article is not clearly stated, it is difficult to find well-written conclusions. The present section of the conclusions contains only a short description of the results obtained, a deep edition of this paragraph is needed.

Author Response

Dear Reviewer,

Thanks for all your comments and suggestions after reviewing our manuscript.

Reviewer’s remark

Authors answer

Comments

English language and style are fine/minor spell check required

Agree

English proof reading was performed repeatedly.

The abstract does not clearly state what the purpose of the study was. Please complete this information very deeply

Agree

The corrections were made in the publication.

Please specify how many schools were included in the survey

Agree

The corrections were made in the publication

Please explain why only students in 1st-4th grade were included in this survey since school lunches are provided in grades 1st-6th?

Don’t agree

According to the regulations of the Cabinet of Ministers No. 614, state-paid school lunches are provided for pupils from 1st to 4th grade.

Please complete this paper with a detailed description of the statistics used and a detailed presentation of the results of the statistical analysis.

Agree

The corrections were made in the publication.

Figure 1. To maintain consistency in the order of the product groups included in the survey and shown in Figure 1, I would suggest reordering the groups in Figure 1.

Agree

The corrections were made in the publication.

Chapter conclusions - line 244 - 246, I would suggest giving examples of better diversification of pupils' healthy diet in line with the recommendations, e.g. more fruit and vegetables etc. In my opinion, this chapter should also briefly mention the general opinion of parents about this food pack solution since finding out this opinion was one of the objectives of this work. For this purpose, it might be worth considering moving the last paragraph from the discussion section to the conclusions?

Agree

The corrections were made in the publication.

The questionnaire should be added to the supplementary materials or to section Materials and Methods.

Don’t agree

The questionnaire is in Latvian, as the research was conducted in the native language of the respondents.

There is no clearly emphasized purpose and sense of the research undertaken, please describe what is the demand for such knowledge, why did the authors decide to take up this topic, please indicate how the obtained results will affect the future behavior of consumers, what will these results bring to science?

Partially agree

The corrections were made in the publication.

The purpose of the study is not to change consumer behaviour, it can be a basis for further research on the nutrition of pupils during the pandemic, how it has affected the eating habits of pupils, BMI, etc.

The present section of the conclusions contains only a short description of the results obtained, a deep edition of this paragraph is needed.

Agree

The corrections were made in the publication.

Reviewer 2 Report

Necessary corrections:

1. The manuscript needs a complete rewrite. The title needs to be more specific, f. ex. "The assessment of school lunches ……in Latvia” (lines: 2-3).

2. The abstract is laconic and could do more to encourage people to familiarize themselves with the paper. The Abstract should consist of an introduction, solid information on the materials and methodology of the study, the results of the study and clear recommendations for the future.

3. In the text are some expressions which need to be up-graded, f. ex. The COVID-19 pandemic (line 10).

4. The text needs to be enrichened by the citations, such as (lines: 28, 34, 50,

5. The text completely lacks of theoretical introduction, which should be presented in the Introduction (after line 63).

6. What do you mean by “all Latvian schools”? (lines: 67-68), The Ministry of Health – of which country? (line: 98).

7. The materials and methods are not explained.

8. There is a lack of hypothesis in the introduction.

9. The hypothesis needs to be veryfied by the results.

10. The hypothesis needs to be accepted / rejected in the conclusions.

11. There is/ are NO research question. Why not? Definitely it / they should occur in order to hepl put the academicians and the readers to follow the main idea of the manuscript.

12. What were the methods used in order to examine the materials and the research sources? The methods need to be described, quoted, and expanded.

13. I urge to expand and enrichen the findings. What do you mean by “The approaches chosen by schools to provide support for the state-funded school lunch varied”? In what ways? Why? Explain, give quotations (see lines: 88-89).

14. Please, highlight the findings. All results, especially those enormous ones need to have very detailed explanation, and thus recommendation to modified it, change it.

15. I suggest to try to take an attempt to make some comparisons between the findings, try to extract the most important findings from the graphic results, and build on it the strength of the findings.

16. It is advisable to give justification to the choices of milk (lines: 177-184), legumes (lines: 185-198), grain products (lines: 199-221).

17. The text needs a strong up-gradation, in the sense of foreign literature. The summary made up out of 20 papers is really not acceptable.

18. In order to enrichen the theoretical introduction, I suggest to cite more references.

Author Response

Dear Reviewer,

Thanks for all your comments and suggestions after reviewing our manuscript.

Reviewer’s remark

Authors answer

Comments

Moderate English changes required

Agree

English proof reading was performed repeatedly.

The manuscript needs a complete rewrite. The title needs to be more specific, f. ex. "The assessment of school lunches ……in Latvia” (lines: 2-3).

Partially agree

The corrections were made in the publication.

The abstract is laconic and could do more to encourage people to familiarize themselves with the paper. The Abstract should consist of an introduction, solid information on the materials and methodology of the study, the results of the study and clear recommendations for the future.

Agree

The corrections were made in the publication.

In the text are some expressions which need to be up-graded, f. ex. The COVID-19 pandemic (line 10)

Don’t agree

The expression (The COVID-19 pandemic) is used in scientific literature, as evidenced by articles in the Scopus database.

The text needs to be enrichened by the citations, such as (lines: 28, 34, 50)

Partially agree

In line 28 and 34, the opinion of the authors is expressed.

A reference has been added to line 50.

The text completely lacks of theoretical introduction, which should be presented in the Introduction (after line 63).

Partially agree

The corrections were made in the publication.

What do you mean by “all Latvian schools”? (lines: 67-68), The Ministry of Health – of which country? (line: 98).

Agree

The corrections were made in the publication.

The materials and methods are not explained.

Agree

The corrections were made in the publication.

There is a lack of hypothesis in the introduction. The hypothesis needs to be veryfied by the results. The hypothesis needs to be accepted / rejected in the conclusions.

Don’t agree

This is not a study that requires a hypothesis. Hypotheses are more often proposed for master's and doctoral theses.

There is/ are NO research question. Why not? Definitely it / they should occur in order to hepl put the academicians and the readers to follow the main idea of the manuscript.

Don’t agree

This is not a study that requires a research question. Research questions are more often proposed for master's and doctoral theses.

What were the methods used in order to examine the materials and the research sources? The methods need to be described, quoted, and expanded

Partially agree

The corrections were made in the publication.

I urge to expand and enrichen the findings. What do you mean by “The approaches chosen by schools to provide support for the state-funded school lunch varied”? In what ways? Why? Explain, give quotations (see lines: 88-89).

Don’t agree

Both the Table 1 and the text below the Table 1 list other approaches.

Please, highlight the findings. All results, especially those enormous ones need to have very detailed explanation, and thus recommendation to modified it, change it.

Agree

The corrections were made in the publication.

I suggest to try to take an attempt to make some comparisons between the findings, try to extract the most important findings from the graphic results, and build on it the strength of the findings.

Partially agree

The corrections were made in the publication.

Reviewer 3 Report

I have read carefully the manuscript entitled "The assessment of school lunches in the form of food packs during the COVID-19 pandemic".

According to the work of manuscript, the idea of the work is good enough to build a wider future assessment. However, this manuscript should improve several points before it will be reconsidered.

Comments:

1.     The aim and particularly novelty of this presented work should be pointed clearly.

2.     The abstract should capture the main points of the paper and highlight the significant obtained results.

3.     In introduction, some crucial and related previous researches of related to similar studies were not included.

4.     The samples strategy collect should be described more clearly.

5.     English should be revised and improved exhaustively throughout the text.

6.     References must be checked again just few examples:

·       Ref. no. 17, pages numbers is missed.

·       Ref. no. 19, check the unity of the ref.

With my regards     

Author Response

Dear Reviewer,

Thanks for all your comments and suggestions after reviewing our manuscript.

Reviewer’s remark

Authors answer

Comments

English language and style are fine/minor spell check required

Agree

English proof reading was performed repeatedly.

The aim and particularly novelty of this presented work should be pointed clearly

Agree

The corrections were made in the publication.

The abstract should capture the main points of the paper and highlight the significant obtained results

Agree

The corrections were made in the publication.

In introduction, some crucial and related previous researches of related to similar studies were not included.

Partially agree

The corrections were made in the publication.

If the reviewer has a specific study in mind that is relevant to this article, please specify.

The samples strategy collect should be described more clearly.

Agree

The corrections were made in the publication.

English should be revised and improved exhaustively throughout the text.

Agree

English proof reading was performed repeatedly.

References must be checked again just few examples:

 Ref. no. 17, pages numbers is missed.

Ref. no. 19, check the unity of the ref.

Partially agree

All references were checked repeatedly.

Ref. no 17 – The publication has no pages. Ref. no. 19 – the publication record is correct.

Round 2

Reviewer 1 Report

Dear Authors,

thank you for your responces. I appreciate your involevment to correct this manuscript, however the construction of ths manuscript as well as the goal of this research still does not bring a new and valuable impact to science world. I wish you all the best on your scientific way.

Author Response

Dear Reviewer,

Thank you for your work in reviewing the publication.

In response to your comment: Your previous suggestions have been taken into consideration. Every researcher has the right to express his opinion, we do not agree with the opinions of the reviewer, moreover, unargued ones.

Reviewer 2 Report

Dear Authors,

I am happy to read the revised version of your article. I see some changes, That is good.

I am pleased to have received an article for review whose authors are from Latvia and have conducted research in Latvia, a country where I have twice received a research grant from the Latvian Ministry of Science and Sports.

The realized research problem is actual and very important, nevertheless, in order for it to meet the requirements of a scientific article (and not a 2-3 page report), it must have a broader context, which will be a comparison of the conducted research by the authors in the international arena.

Unfortunately, in my opinion, this article describing the results of a study does not meet the requirements of an article, because the researcher did not confront the problem in the widespread, broad dimension. With all due respect, it is not that we have many countries in Europe boasting a population of a few million people, but that any study is worth showing in a broader, that is, macro or meso perspective. I'm already hinting, in the case of this article, you need to review in other countries, or in Europe, or in the world, and what's more, show various recommendations, which will certainly be a huge added value for this study. 

I wish the authors all the best, I enjoy this study, and I hope for your understanding and look forward to further improvements in the text.

Best regards

your diligent reviewer

Author Response

Response to Reviewer 2.

Dear Reviewer,

Thank you for your work in reviewing the publication.

In response to your comment: Your previous suggestions have been taken into consideration. I agree that it is important to evaluate other articles when possible, but for this article there are several arguments for not doing so:

  1. Schools in most European countries do not provide state-paid lunches.
  2. Scientific articles on school lunches are scarce, which is related to the above argument. There are relatively many articles about the US school lunch program, but it is not really comparable to Latvia.
  3. Scientific articles about the Covid-19 pandemic are related to changing eating habits, increasing BMI, poverty, etc., which is not really related to the purpose of this article.

I would like to clarify the name of the ministry mentioned in the review – Ministry of Education and Science, it would be correct to indicate the correct name of the ministry if you have received grants from this ministry twice.

Reviewer 3 Report

Authors have addressed all of the comments

Author Response

Dear Reviewer,

Thank you for your work in reviewing the publication.